# Unraveling the Molecular Basis of the Dystrophic Process in Limb-Girdle Muscular Dystrophy LGMD-R12 by Differential Gene Expression Profiles in Diseased and Healthy Muscles

**DOI:** 10.3390/cells11091508

**Published:** 2022-04-30

**Authors:** Christophe E. Depuydt, Veerle Goosens, Rekin’s Janky, Ann D’Hondt, Jan L. De Bleecker, Nathalie Noppe, Stefaan Derveaux, Dietmar R. Thal, Kristl G. Claeys

**Affiliations:** 1Laboratory for Muscle Diseases and Neuropathies, Department of Neurosciences, KU Leuven, and Leuven Brain Institute (LBI), Herestraat 49, 3000 Leuven, Belgium; christophe.depuydt@kuleuven.be; 2Department of Radiology, University Hospitals Leuven, Herestraat 49, 3000 Leuven, Belgium; veerle.goosens@uzleuven.be (V.G.); nathalie.noppe@uzleuven.be (N.N.); 3VIB Nucleomics Core, Herestraat 49, 3000 Leuven, Belgium; rekins.janky@vib.be (R.J.); stefaan.derveaux@vib.be (S.D.); 4Department of Neurology, University Hospitals Leuven, Herestraat 49, 3000 Leuven, Belgium; ann.dhondt@uzleuven.be; 5Department of Neurology, University Hospital Gent, Corneel Heymanslaan 10, 9000 Gent, Belgium; jan.debleecker@ugent.be; 6Department of Pathology, University Hospitals Leuven, Herestraat 49, 3000 Leuven, Belgium; dietmar.thal@kuleuven.be; 7Laboratory for Neuropathology, Department of Imaging and Pathology, KU Leuven, and Leuven Brain Institute (LBI), Herestraat 49, 3000 Leuven, Belgium

**Keywords:** anoctamin-5, *ANO5*, LGMD2L, muscle dystrophy, RNA-seq, transcriptomics, gene signatures, selective muscle involvement, muscle biopsy, fibroadipogenic progenitors

## Abstract

Limb-girdle muscular dystrophy R12 (LGMD-R12) is caused by two mutations in anoctamin-5 (*ANO5*). Our aim was to identify genes and pathways that underlie LGMD-R12 and explain differences in the molecular predisposition and susceptibility between three thigh muscles that are severely (semimembranosus), moderately (vastus lateralis) or mildly (rectus femoris) affected in this disease. We performed transcriptomics on these three muscles in 16 male LGMD-R12 patients and 15 age-matched male controls. Our results showed that LGMD-R12 dystrophic muscle is associated with the expression of genes indicative of fibroblast and adipocyte replacement, such as fibroadipogenic progenitors and immune cell infiltration, while muscle protein synthesis and metabolism were downregulated. Muscle degeneration was associated with an increase in genes involved in muscle injury and inflammation, and muscle repair/regeneration. Baseline differences between muscles in healthy individuals indicated that muscles that are the most affected by LGMD-R12 have the lowest expression of transcription factor networks involved in muscle (re)generation and satellite stem cell activation. Instead, they show relative high levels of fetal/embryonic myosins, all together indicating that muscles differ in their baseline regenerative potential. To conclude, we profiled the gene expression landscape in LGMD-R12, identified baseline differences in expression levels between differently affected muscles and characterized disease-associated changes.

## 1. Introduction

The limb-girdle muscular dystrophies (LGMDs) constitute a group of rare, progressive and genetic muscle disorders, with weakness and atrophy of mainly pelvic and shoulder girdle muscles [1]. LGMDs are inherited mostly in an autosomal recessive manner (LGMD-R), more frequent than autosomal dominant (LGMD-D). Autosomal recessive pathogenic variants in the anoctamin-5 encoding gene (*ANO5*) cause LGMD-R12 anoctamin5 related (LGMD-R12) and distal Miyoshi muscular dystrophy type 3 (MMD3) [2,3,4]. In *ANO5*-related muscular dystrophies, a male predominance and women often showing a less severe phenotype have been reported [4]. Dominant *ANO5* mutations cause the bone disorder gnathodiaphyseal dysplasia 1 (GDD1) [5,6]. To date, pathophysiology of *ANO5*-related muscular dystrophies is largely unknown and disease-specific treatments do not exist.

The *ANO5*-gene is highly expressed in skeletal and cardiac muscles and in bones [6]. ANO5 or transmembrane 16E protein (ANO5/TMEM16E) belongs to a family of ten transmembrane proteins that have a role either as calcium-activated ion channels, lipid scramblases, or both [7,8]. ANO5 is the only member of this protein family associated with muscular dystrophy. Its lipid scramblase and ion channel activities have been shown to play a role in sarcolemmal repair and myoblast fusion during muscle regeneration [9,10,11,12,13].

Similar to other muscular dystrophies, the muscle biopsy in LGMD-R12 reveals dystrophic changes consisting of necrotic and regenerating muscle fibers, and replacement of muscle cells by connective and fatty tissue as disease progresses [3,14]. In addition, muscle biopsy in LGMD-R12 often reveals inflammatory infiltrates [15,16,17]. Although pathomechanisms of the dystrophic process in LGMD-R12 and other muscular dystrophies are largely unknown, several pathways that might contribute to progressive muscle cell death have been suggested, such as oxidative stress [18], mitochondrial dysfunction [19], defective membrane repair [9], inflammation [20], compromised regeneration [13] and impaired satellite cell activation [21].

Interestingly, although the genetic defect is the same in all muscles in LGMD-R12, specific muscles are selectively involved, which can be seen on muscle Magnetic Resonance Imaging (MRI). LGMD-R12 patients show a predominant affection of the posterior thigh muscles, starting at the semimembranosus, whereas vastus lateralis is affected later in the disease course, and rectus femoris, gracilis, sartorius and biceps femoris short head show fatty infiltration only at more advanced stages of the disease [4,22,23]. A selective muscle involvement also occurs in other muscular dystrophies, but the patterns of affected muscles differ depending on the mutated gene [23]. It is highly relevant from a therapeutic perspective to understand the reasons why some muscles escape the dystrophic effects of disease-causing mutations.

The degree of muscle involvement can be determined semiquantitatively on muscle MRI using various scales, such as the 4-point Mercuri score, which has been shown to correlate with histopathological findings and disease progress [24,25].

The objectives of this study were to identify genes and pathways that underlie LGMD-R12, resulting in new molecular insights into the dystrophic process, and explain differences in the molecular predisposition and susceptibility between different muscles. We used RNA-seq and studied the gene expression profiles in three differentially affected muscles of the thigh (severely affected semimembranosus, intermediately affected vastus lateralis, mildly affected to preserved rectus femoris) in 16 male LGMD-R12 patients and in the same three muscles in 15 healthy male age-matched control individuals. We identified marker genes and pathways for the different muscles in patients and in controls and applied single cell integration and deconvolution analysis to estimate how disease affects different cell types that compose human skeletal muscle tissue.

## 2. Patients and Methods

### 2.1. Patients and Controls

We included 16 symptomatic, ambulatory, adult, male patients with genetically proven LGMD-R12, i.e., carrying two pathogenic variants in the anoctamin-5 gene (*ANO5*) (Table 1), and 15 age-matched healthy male control individuals. Patients were followed at the Neurology Department, Neuromuscular Reference Center, at the University Hospitals Leuven and Gent, Belgium. Wheelchair-bound patients and/or patients with complete fatty infiltration of the three target muscles (semimembranosus, vastus lateralis, rectus femoris) on muscle MRI were excluded, as well as patients with a predominantly distal Miyoshi muscular dystrophy type 3 (MMD3) phenotype or with asymptomatic hyperCKemia. Further exclusion criteria were: presence of a contra-indication for MRI (e.g., MRI non-compatible pacemaker or claustrophobia), a blood coagulation disorder, use of anticoagulation medication or potentially toxic medication on muscles such as steroids or statins at the time of muscle biopsy and active alcohol abuse.

In all participants, the 6-min walk distance (6MWD) and 10-m walk test (10MWT) were performed (Table 1) and a whole-body muscle MRI (1.5 Tesla, Philips Ingenia, Philips Medical Systems, Best, The Netherlands) with axial and coronal T1-weighted scans was carried out, prior to muscle biopsy sampling. The target muscles were scored on the MR images using the Mercuri score, which is a 4-point grading system to categorize disease severity in individual muscles based on visual inspection of fatty tissue infiltration on MRI: normal appearance (score 1/normal), less than 30% affected (score 2/mild), between 30–60% affected (score 3/moderate), between 60–100% affected (score 4/severe) [24].

The study was approved by the Ethics Committee Research UZ/KU Leuven (S-59867). Written informed consent was obtained from all participants.

### 2.2. Muscle Biopsies

In all patients and controls, three muscle biopsies were taken, one in the semimembranosus (SM), one in the vastus lateralis (VL) and one in the rectus femoris (RF) muscle. We performed vacuum-assisted needle biopsies using the EnCor Enspire (breast) biopsy system with 10G EnCor needles (Bard Benelux, Olen, Belgium). For every muscle biopsy a separate needle was used. Muscle biopsies were taken under ultrasound guidance after injecting local anesthesia (lidocaine 2.0%) in the skin, subcutaneous fat and fascia but not in the muscle itself. Prior to the muscle biopsy procedure, a whole-body muscle MRI including the thigh muscles was carried out to determine the level for biopsy sampling, in order to avoid completely fatty replaced tissue. Since the muscle transcriptome is known to be altered by several variables, the following measures were taken, as well as matching for age and sex, to eliminate possible confounding factors: biopsies were taken in the morning, to avoid differences in circadian rhythm between participants [26], and participants were instructed not to perform intensive physical activities for at least one week before the muscle sampling, in order to avoid bias from differences in exercise or work load [27,28].

Immediately after sampling, the biopsies from each of the three muscles were mounted on a separate cork and snap frozen in isopentane that was cooled with liquid nitrogen. All biopsies were stored at −80 °C. For RNA extraction, 200 slices of 5 µm thickness from each biopsy were cut with a microtome-cryostat and collected in two RNase-free tubes of 2.0 mL (100 slices in each tube), which were kept frozen at −80 °C until RNA extraction was performed.

### 2.3. RNA Procedures

#### 2.3.1. RNA Extraction

We used the same protocol for RNA extraction from muscle tissue as the one described by Cummings et al. [29]. RNA was extracted from the 93 muscle biopsies using the Direct-zol RNA Miniprep Plus kit ZY-R2073 (Zymo Research, Irvine, CA, USA) according to the manufacturer’s instructions. In order to avoid technical variability, all samples were extracted with the same kit on two consecutive days in the same lab by the same lab technician.

#### 2.3.2. RNA Quality Control

The 93 extracted RNA samples were treated with DNase prior to their submission to the VIB Nucleomics Core (Leuven, Belgium, www.nucleomics.be, accessed on 15 July 2019). The Nanodrop ND-1000 (Nanodrop Technologies, Wilmington, DE, USA) was used to measure RNA concentration and purity spectrophotometrically. A Bioanalyzer 2100 (Agilent, Santa Clara, CA, USA) was applied to assess RNA integrity. The RNA Integrity Number (RIN) scores mostly varied from 6.3 (partially degraded) to 8.4, except for a few samples that were highly degraded.

#### 2.3.3. Library Preparation

Per sample, an amount of 5 ng of total RNA (or 7 µL RNA when 5 ng was not available) was used as input for the SMART-Seq Stranded Kit (Cat. No. 634444; protocol version “022819”; low input option; Takara Bio USA, San Jose, CA, USA). This kit can deal with degraded as well as high-integrity input RNA. Positive and negative controls were included in the experimental design using 5 ng control RNA (included in the kit) and 7 µL RNase-free water, respectively.

First, RNA is converted to cDNA using random priming (scN6 Primer) and SMART (Switching Mechanism At 5′ end of RNA Template) technology (Takara Bio USA, San Jose, CA, USA) and then full-length adapters for Illumina sequencing (including specific barcodes for dual-indexing libraries) are added through PCR using a limited number of cycles (5 cycles). The PCR products are purified and then ribosomal cDNA is selectively depleted by cleaving the ribosomal cDNAs by scZapR in the presence of mammalian-specific scR-Probes, which target nuclear and mitochondrial rRNA sequences. The library fragments derived from non-rRNA molecules remain untouched by this process. The remaining cDNA fragments are further amplified with primers universal to all libraries (14 cycles). Lastly, the PCR products are purified once more to yield the final cDNA library. All libraries were finally quantified using Qubit dsDNA HS kit (Thermo Fisher Scientific, Waltham, MA, USA) and their size distribution was checked using a Bioanalyzer 2100 (Agilent, Santa Clara, CA, USA). Nine of the 93 libraries were excluded from downstream sequencing as there was no yield: control C6 (VL), control C8 (SM), patient 6 (SM), patient 7 (SM), patient 8 (VL), patient 12 (RF) and patient 16 (SM, VL, RF) (Appendix A).

#### 2.3.4. Sequencing (RNA-Seq)

Sequence-libraries of each of the 84 remaining samples were equimolarly pooled and sequenced on two Illumina NovaSeq 6000 S1 100 flow-cells (Xp workflow, Paired Read 51-8-8-51, 1% PhiX v3) (Illumina, San Diego, CA, USA) at the VIB Nucleomics Core (Leuven, Belgium, www.nucleomics.be, accessed on 12 September 2019). RNA-seq was performed twice, in two runs. In the first run, data from 84 samples were analyzed (i.e., 93 samples minus 9 excluded samples because of too low RNA amount). The first run was successful with high quality (~95% of the bases ≥30) and output according to the expectations (~2000 M Passed Filter clusters). With 25M PF reads on average per sample, the output per sample was quite variable, ranging from 9.4M to 47.4M PF reads. We also observed a high percentage of duplicates in some samples and a high percentage of adapters in the sequences, corresponding to samples with very low input material. The mapping of the reads is relatively correct (90%), but the filtering based on the mapping quality has a huge impact (~50% loss of reads for many samples), which should be due to duplicated reads as the filtering discards similar reads. Out of 84 samples, we selected the 38 best samples (<22% No Features, Number of final counts > 12M), whereas 46 samples were pooled for a second sequencing run in order to reach 20M final counts (=assigned reads) for each sample.

### 2.4. Data Analysis

#### 2.4.1. Preprocessing

Using FastX 0.0.14 and Cutadapt 1.15 (http://hannonlab.cshl.edu/fastx_toolkit/index.html, accessed on 20 September 2019), low quality ends and adapter sequences were trimmed off from the Illumina reads. Next, the following were filtered using FastX 0.0.14 and ShortRead 1.36.1 [30]: small reads (length < 35 bp), polyA-reads (more than 90% of the bases equal A), ambiguous reads (containing N), low-quality reads (more than 50% of the bases < Q25) and artifact reads (all but three bases in the read equal one base type). We then identified and removed reads that align to phix_illumina applying Bowtie2 2.3.3.1 [31].

#### 2.4.2. Mapping

We aligned the preprocessed reads to the reference genome of *Homo sapiens* (GRCh38) with STAR aligner v2.5.2b [32]. Default STAR aligner parameter settings were used, except for “—outSAMprimaryFlag OneBestScore—twopassMode Basic—alignIntronMin 50–alignIntronMax 500,000—outSAMtype BAM SortedByCoordinate”. Reads with a mapping quality smaller than 20 were removed from the alignments using Samtools 1.5 [33].

#### 2.4.3. Counting

We counted the number of reads in the alignments that overlap with gene features using featureCounts 1.5.3 [34]. We chose the following parameters: -Q 0 -s 2 -t exon -g gene_id, and removed genes for which all samples had less than 1 count per million. We further corrected raw counts within samples for GC-content and between samples using full quantile normalization, with the EDASeq package from Bioconductor [35].

### 2.5. RNA-Seq Analysis

Raw count matrices were uploaded to the online R-based UniApp data analysis platform (Unicle Biomedical Data Science, Leuven, Belgium) for analysis. We used the R packages plotly and ggplot to generate bar, pie, line and scatter plots.

#### 2.5.1. Quality Control and Data Normalization

Genes expressed at a level of at least 1 count per million reads in at least 10% of samples were filtered and normalized using the EdgeR package. Samples with less than 1 Million reads were filtered out from the final analysis.

#### 2.5.2. Principal Component Analysis (PCA)

We first normalized the raw count RNA-seq data from 84 patients for total read depth. Next, we performed quality filtering for read mapping and the number of detected genes and excluded 5 samples: patient 3 (SM), patient 5 (SM, VL), patient 12 (SM, VL) (Appendix A). The normalized data were auto-scaled and PCA was performed on the top 2000 most highly variable genes (with the Seurat R package [36]) to build a 2-dimensional representation of the data.

#### 2.5.3. Pair-Wise Differential Analysis

We used limma [37] for differential expression analysis between two specific clusters and visualized using a volcano plot.

#### 2.5.4. Gene Set Enrichment Analysis (GSEA)

We applied gene set enrichment analysis (clusterProfiler R package) to compare gene expression profiles between two groups [38]. We performed GSEA using a subset of gene sets selected from the Molecular Signatures Database (MSigDB version 7.41; http://bioinf.wehi.edu.au/software/MSigDB/, accessed on 23 September 2021), which is a collection of annotated gene sets. GSEA scores were calculated for sets with at least five detected genes, all other parameters were considered default.

#### 2.5.5. Heatmap Analysis

Gene expression heatmaps are based on averaged auto-scaled data. We produced heatmaps by the heatmaply R package.

#### 2.5.6. Deconvolution Analysis

Bulk RNA-seq count data were deconvoluted into cell composition matrices with the MUSIC algorithm [39] on a reference single cell RNA-seq dataset derived from the mononuclear cells of the human vastus lateralis [40] with default parameters.

### 2.6. Single Cell scRNA-Seq Analysis

We used a publicly available single cell dataset of human muscle to obtain markers for different stromal cell populations. The raw count matrix of human muscle cells as described in the study “Single cell transcriptional profiles in human skeletal muscle” was downloaded from the GEO repository under the accession code GSE130646 [40].

#### 2.6.1. Quality Control and Data Normalization

We applied the following quality control steps for the human freshly-isolated muscle cells: (i) genes expressed by <3 cells were not considered; (ii) cells that had over 2000 expressed genes (possible doublets), or over 4% of unique molecular identifiers (UMIs) derived from the mitochondrial genome were deleted. The data of the remaining cells were natural-log transformed using log1 *p* and normalized using the Seurat package [36].

#### 2.6.2. Dimension Reduction

After auto-scaling, genes with high variability were identified using the Seurat FindVariableGenes function. The top 2000 most highly variable genes were included in the analysis. Subsequently, the normalized data were first summarized by PCA, and the first 20 PCAs were visualized using t-Distributed Stochastic Neighbor Embedding (t-SNE, Rtsne package) with a perplexity value of 100 and a learning rate of 100. This representation was only used to visualize the data.

#### 2.6.3. Clustering Analysis and Annotation

We performed graph-based clustering to cluster cells according to their gene expression profiles as implemented in Seurat with the first 20 PCA dimensions, number of neighbors set to 10 and at a resolution of 0.8. Cell clusters were annotated based on canonical markers.

#### 2.6.4. Pair-Wise Differential Analysis

We performed differential expression analysis between two specific clusters using limma [37].

#### 2.6.5. Marker Gene Analysis

A two-step approach was used to obtain ranked marker gene lists for each cluster. Marker genes for a given cluster should have the highest expression in that cluster compared to all other clusters and are therefore uniquely assigned to one cluster. Next, marker genes were ranked using a product-based meta-analysis [41]. We performed pair-wise differential analysis of all clusters against all other clusters separately and ranked the results of each pair-wise comparison by log2 fold change. The most downregulated genes received the highest rank number and the most upregulated genes received the lowest rank number (top ranking marker genes). For each cluster, we combined the rank numbers for all genes in all pair-wise comparisons by calculating their product to obtain a final list of ranked marker genes for each cluster. Clusters were annotated based on literature-curated marker genes of canonical muscle cells. Cells that could not be unambiguously assigned to a biologically meaningful phenotype were excluded from the analysis because they might represent low quality cells or doublets.

#### 2.6.6. Heatmap Analysis

To account for cell-to-cell transcriptomic stochastics, all heatmaps are based on cluster-averaged gene expression. For visualization, data were auto-scaled. Using the heatmaply package, heatmaps were produced.

## 3. Results

### 3.1. Demographic Data

Mean age at inclusion was 45.7 years (range 26 to 64 years) for the patients (Table 1) and 45.1 years (range 25 to 63 years) for the controls (*p* > 0.05). Further clinical and genetic data of the patients are detailed in Table 1.

### 3.2. Mercuri Score Is Correlated with Functional Outcomes in LGMD-R12 Patients

We used the 4-point Mercuri score to categorize disease severity in the three target muscles (SM, VL, RF) based on visual inspection of fatty tissue infiltration on muscle MRI in all patients and controls [24]. Mercuri scores in patients’ muscles varied between 1 and 4 (Table 1, Figure 1A), whereas in controls the Mercuri score for all muscles was equal to 1 (i.e., normal) (Figure 1B). The Mercuri score in the patients was lowest for the rectus femoris with mean 1.2 (95% CI 1.0–1.4) (least affected), intermediate for the vastus lateralis with mean 2.0 (95% CI 1.3–2.7) (intermediately affected) and highest for the semimembranosus with mean 2.8 (95% CI 2.1–3.4) (most affected) (ANOVA, *p* = 0.001) (Figure 1C), which is consistent with the literature [4,22,23]. We calculated the Mercuri score with years after disease onset and showed that the Mercuri score significantly increased over time with increasing disease duration (Pearson correlation coefficient r = 0.5666, *p* = 0.0331) (Figure 1D). Two patients (Table 1, patients 7 and 11) showed a longer disease duration with, however, lower Mercuri scores (Figure 1D), which might be explained by the different genetic defects in *ANO5* of these patients compared to the others (Table 1). Functionally, the Mercuri score significantly increased with decreasing 6MWD (r = −0.8387, *p* < 0.001; Figure 1E) and with increasing 10MWT (r = 0.8543, *p* < 0.001; Figure 1F), which corresponds to the progressive functional decline of the disease over time.

### 3.3. Unbiased Analysis: Gene Expression Profiles Correlate with the Mercuri Score

We performed principal component analysis (PCA) using highly variable genes as input and correlated gene expression signatures with the Mercuri score (Figure 2). Interestingly, color coding the unbiased PCA plot for the Mercuri score showed that the first component (the one that explains most variability in the data and hence is the strongest signature) was highly correlated with the Mercuri score. The second component was correlated with the mitochondrial read count (Figure 2). To determine whether the gene signature that correlated with the Mercuri score was based on genes involved in muscle fiber degradation and connective and fat tissue deposition, we color coded the PCA plot for expression of a known muscle membrane gene (alpha-sarcoglycan, *SGCA*) [42], fibroblast marker (collagen I A1, *COL1A1*) [43] and adipocyte marker (adiponectin, *ADIPOQ*) [44]. This analysis showed a gradient of decreasing expression of muscle marker genes and a gradual increase in the expression of fibroblast, adipocyte and other stromal cells (Figure 2). Thus, our exploratory PCA analysis was consistent with the replacement of muscle tissue by connective and fat tissue as disease progresses.

### 3.4. Quantification of Gene Signatures Associated with LGMD-R12

We next aimed to formally quantify changes in gene expression signatures, by performing a differential gene expression analysis of LGMD-R12 patients versus controls (irrespective of the Mercuri score, muscle or other stratification criteria) (Figure 3A). This analysis showed in patients an increase in genes involved in response to muscle injury and inflammation, such as osteopontin (*SPP1*) [45,46], immunoglobulin heavy constant gamma 1/2 (*IGHG1/2*), leukocyte receptor tyrosine kinase (*LTK*), joining chain of multimeric IgA and IgM (*JCHAIN*), serum amyloid A1 (*SAA1*), as well as embryonic and fetal myosins, including myosin heavy chain 3 (*MYH3*) and myosin light chain 4 (*MYL4*). Embryonic myosins are normally not expressed in adult muscle but can be upregulated in response to injury in regenerating muscle fibers [47]. Interestingly, the LGMD-R12 disease-causing gene *ANO5* ranked in the top 0.1% of most downregulated genes, whereas other anoctamin genes showed a trend to being slightly upregulated (*ANO3, ANO7, ANO9, ANO7L1*) (Figure 3A). These findings correspond to recent hypotheses that anoctamin homologs might compensate for the loss of *ANO5* in mice [48].

To quantify gene signature changes at the gene set level we performed a gene set enrichment analysis (GSEA) of the hallmark gene sets based on the results of the differential gene expression analysis of LGMD-R12 patients versus controls (Figure 3B). This analysis showed that LGMD-R12 patients have upregulated gene sets related to immune response, phagocytosis and collagen remodeling while having downregulated gene sets relating to energy metabolism and protein synthesis.

### 3.5. Pathway Analysis Characterizes Gene Sets for LGMD-R12 Progression

To characterize gene sets for LGMD-R12 progression we performed GSVA analysis to alter the gene-by-sample matrix to a gene set-by-sample matrix and used the latter matrix to perform a differential analysis between semimembranosus muscle samples with Mercuri score 1 versus Mercuri score 4 (Figure 3C,D) [49]. This calculation showed that the cell type signature of skeletal muscle cells was downregulated in Mercuri score 4 samples, while stromal and immune cell signatures were upregulated. Consistently, when evaluating cellular pathways in a similar analysis, we found that pathways associated with skeletal muscle were downregulated (ribosomes, mitochondria and aerobic respiration), while gene sets associated with cell death, inflammation and collagen metabolism were upregulated (Figure 3C,D). To evaluate gradual differences in pathway expression between different Mercuri scores, we performed marker pathway analysis on the GSVA transformed matrix and visualized the results as a heatmap. This assessment showed a progressive downregulation of pathways such as myogenesis, protein metabolism and oxidative phosphorylation, while pathways associated with inflammation and cell death were upregulated (Figure 3E).

### 3.6. Deconvolution Analysis Shows an Increase in Fibroadipogenic Progenitor (FAP) Cells in LGMD-R12 Muscles

Our analysis of pathways associated with LGMD-R12 were consistent with a decrease in muscle and an increase in stromal cell signatures. However, this analysis did not provide insight into whether these findings reflect an overall change in gene expression in all cells of the muscle, or if the actual cellular composition of the disease muscle is changed. To estimate differences in cell type composition we performed a deconvolution analysis [50]. For this analysis we used marker genes derived from a recently described skeletal muscle single cell RNA-seq dataset, cell types that together make up human skeletal muscle tissue (Appendix A) [40], and deconvoluted our bulk RNA-seq dataset. While preliminary, our analysis indicated a progressive increase in FAP lumican+ (LUM+) cells and their signatures but a decrease in (vascular) smooth muscle signatures (vSMC) (Figure 3F,G).

### 3.7. Distinct Gene Signatures in Different Muscles in Healthy Controls

To investigate the hypothesis that baseline differences in the molecular composition of different muscles influence susceptibility to muscular dystrophy, we performed a detailed analysis of gene signatures of the three biopsied muscles in healthy control individuals. We first performed a PCA analysis, which showed a clear separation of semimembranosus, vastus lateralis and rectus femoris on the third component (this first component highlighted a lowly sequenced outlier sample) (Figure 4A). Color coding PCA plots for marker genes for fast- and slow-twitch muscle showed a clear gradient, with the rectus femoris showing the highest expression of the fast-twitch muscle myosins (myosin heavy chain 4 (*MYH4*), myosin heavy chain 2 (*MYH2*), myosin heavy chain 1 (*MYH1*) and myosin light chain 1 (*MYL1*)) and the semimembranosus presenting the highest expression of the slow-twitch markers (myosin heavy chain 7 (*MYH7*), myosin heavy chain 6 (*MYH6*) and myosin light chain 3 (*MYL3*)) (Figure 4B,C) [51,52]. The vastus lateralis expressed intermediate levels of fast- and slow-twitch myosins. Differential analysis and gene set analysis between the most affected (semimembranosus) and less affected (rectus femoris) muscles showed differences in genes that regulated anterior–posterior axis planning, fast- and slow-twitch muscle fibers and satellite cell marker genes (Figure 4D,E).

### 3.8. Differential Gene Expression between Different Muscles: Identification of Genes Associated with Different Muscles

To unbiasedly assess genes enriched in the different biopsied muscles (SM, VL, RF) we performed a marker gene analysis (Figure 4F–H; Table 2 and Table 3). In this analysis, each muscle is compared to the other muscles separately via a differential gene expression analysis. The results of the different analyses are then combined with a product-based meta-analysis based on the log-fold changes [53]. We used a heatmap to visualize the top 20 marker genes. Interestingly, several transcription factors were highly differentially expressed across semimembranosus, such as Heart And Neural Crest Derivatives Expressed 2 (*HAND2*), HAND2 Antisense RNA 1 (*HAND2-AS1*), Homeobox D8 (*HOXD8*) and Homeobox D9 (*HOXD9*), vastus lateralis, such as T-Box Transcription Factor 5 (*TBX5*), Homeobox A13 (*HOXA13*), T-Box Transcription Factor 5 Antisense RNA 1 (*TBX5-AS1*), and rectus femoris such as Transcription Factor 24 (*TCF24*) (Figure 4F). In addition, we identified differential expression of genes involved in the regulation of protein synthesis in muscle tissue (*METTL21C*) [54] and glycogen breakdown were highest in the rectus femoris, while adipogenic and lipid pathways were upregulated in the semimembranosus (Figure 4F–H). A similar analysis of early stage (Mercuri score 1 and 2) and late stage (Mercuri score 3 and 4) muscles showed that most variation in disease muscle was explained by disease processes (Appendix A). Together, these findings may suggest that muscles differ in their baseline regenerative potential.

## 4. Discussion

The main objective of this study was to identify genes and pathways that underlie LGMD-R12 and explain differences in the molecular predisposition and susceptibility between three different thigh muscles that are severely (SM), moderately (VL), or mildly (RF) affected in this disease. We performed an unbiased analysis of 84 muscle biopsies (79 after quality checks) of these three differently affected muscles in 16 LGMD-R12 patients and 15 age-matched controls (all males). Our results indicated that, at the cellular and molecular level, LGMD-R12 muscular dystrophy is characterized by the expression of genes indicative of fibroblast and adipocyte replacement, such as fibroadipogenic progenitor (FAP) cells and immune cell infiltration, while gene signatures associated with striated muscle (protein synthesis, OXPHOS, glycogen-, glucose- and amino acid metabolism) are downregulated in dystrophic muscle. Furthermore, muscle degeneration as quantified by the radiological Mercuri score is associated with an increase in genes associated with muscle injury and inflammation, as well as genes involved in muscle repair/regeneration (including embryonic and fetal myosins). We also identified interesting differential expression patterns in other anoctamin genes that are upregulated in response to *ANO5* loss. Analysis of baseline differences in between muscles in healthy individuals indicated that muscles that are the most affected by LGMD-R12 have the lowest expression of transcription factor networks involved in muscle (re)generation and satellite (stem) cell activation. Instead, they show relatively high levels of fetal/embryonic myosins. This is the first study on transcriptomics in LGMD-R12 and the first study overall in which three different muscles were biopsied in patients and in healthy controls, enabling us to conclude to these important and highly relevant findings.

### 4.1. Genes Involved in Inflammation Are Upregulated in Dystrophic LGMD-R12 Muscles

Our study showed an increase in genes involved in muscle injury and inflammation in dystrophic muscles of LGMD-R12 patients. The number of different immune cells is largely increased compared with healthy muscle. This has been reported in muscles in Duchenne muscular dystrophy (DMD) patients and mdx-mice, including T-lymphocytes (CD4, CD8 and regulatory T-cells or TRegs), natural killer (NK) cells, neutrophils, eosinophils and macrophages [20,55,56]. Furthermore, histological analyses of muscle biopsies from LGMD-R12 patients often show inflammatory changes, characterized by CD45 and CD8 positive leukocytes as well as CD68 and CD206 positive macrophages accumulating within myofibers showing myophagocytosis [16,17]. Inflammatory changes have been reported in the muscle biopsies of patients with other muscular dystrophies as well, such as DMD [57], facioscapulohumeral dystrophy (FSHD) [58], LGMD-R2 [59], LGMD R3 and LGMD R5 [60]. Moreover, on muscle MRI, there are indications that inflammation plays a role in muscular dystrophies, such as LGMD-R12 [16,17] and FSHD [61,62]. Further evidence comes from animal models such as the ANO5-knock-out rabbit model, where scattered necrotic muscle fibers with inflammatory infiltrates are observed in muscle tissue [48].

The precise role of inflammation in the pathomechanism of muscular dystrophies is not well understood. Moreover, it is unclear if the inflammation is rather a primary feature or secondary to muscle degeneration. In contrast to acute muscle injury, the continuing injuries in dystrophic muscle result in the permanent recruitment of pro-inflammatory monocytes and the presence of pro-inflammatory macrophages in the muscle, leading to chronic inflammation [63]. Because of the asynchronicity of the injuries and of the regenerative cues, macrophages adopt a mixed phenotype making them less efficient in myogenesis, while they stimulate fibroadipogenic (FAP) cells (see below) and fibroblastic cells to produce extracellular matrix, leading to fat infiltrates and fibrosis. Chronic inflammation further worsens the disease, and can thus be considered as a secondary event in muscular dystrophies [55,56,64]. Therapeutic strategies inducing a change in macrophages that are present in the dystrophic muscle towards an anti-inflammatory profile could be beneficial, as has been shown in mdx-mice [64]. Another example of therapeutic intervention could be the depletion of osteopontin, which is an immunomodulator that is highly expressed in dystrophic muscles, as was shown in LGMD-R12 muscular dystrophy in our study. A recent study showed that osteopontin ablation ameliorated muscular dystrophy by shifting macrophages to a pro-regenerative phenotype in DMD [45].

In addition, our results showed an increased expression of serum amyloid A1 (*SAA1*) in dystrophic muscles in LGMD-R12 patients. Serum amyloid A protein (SAA) is an acute-phase protein, which is normally soluble and shows the highest concentration in plasma during inflammation, but which can be abnormally deposited in AA amyloidosis, secondary to chronic inflammatory processes, as fibers of insoluble protein in the extracellular space of various organs and tissues. Interestingly, muscle biopsies of some LGMD-R12 patients showed amyloid depositions around muscle fibers and within blood vessel walls [65,66]. Amyloid subtyping showed the presence of apoliprotein E, apoliprotein A1, apoliprotein A4, serum Amyloid P component and gelsolin in the deposits, whereas SAA was not typed in the depositions by Liewluck et al. [66]. Perhaps the technology was not sensitive enough to detect SAA in the amyloid deposits, or its absence could be explained by molecular interactions between amyloid precursors and associated conformational changes [67].

### 4.2. Fibroadipogenic Progenitor (FAP) Cells Are Upregulated in LGMD-R12 Muscles

Our data showed that fibroadipogenic progenitor cells (FAPs) are most highly expressed in the more severely affected muscles of LGMD-R12 patients. FAPs are muscle interstitial mesenchymal cells that interact with myogenic stem cells (satellite cells) to support myogenesis and muscle regeneration [68,69,70]. FAPs also have a role in muscle fibrosis and fatty tissue replacement, as is seen in muscular dystrophies [21,69,71,72]. FAPs undergo a large expansion, followed by their macrophage-mediated clearance and the reestablishment of their steady-state pool, during the first days following muscle injury. During this exact time window, FAPs establish a dynamic network of interactions via chemokine and interleukin signaling that culminate in muscle repair, together with the other cellular components of the muscle stem cell niche [73]. In muscular dystrophies, where dystrophic myofibers undergo repeated rounds of injury, the autocrine/paracrine constraints controlling FAP adipogenesis are released, which leads to fat infiltrates [74]. Duchenne muscular dystrophy (DMD) patients and mdx-mouse models show increased expression of platelet-derived growth factor receptor alpha (PDGFRα), which is an FAP marker. Furthermore, FAP accumulation correlates with increased fibro-fatty degeneration and disease severity [70,75]. High transforming growth factor beta (TGF-β) is an inflammatory factor produced in the dystrophic muscle that contributes to increased FAP accumulation and differentiation into fibrogenic and adipogenic fates [72,75]. Similar FAP involvement has also been shown in other muscular dystrophies, such as limb-girdle muscular dystrophy R2 (LGMD-R2) [76] and facioscapulohumeral muscular dystrophy (FSHD) [62,77]. Several clinical trials evaluating the effects of different drugs that alter FAP fate are being/were performed in DMD and LGMD-R2 [21]. Our deconvolution analysis suggested that, specifically, the recently described FAP lumican-positive (LUM+) subpopulation might be involved in the pathogenesis of LGMD-R12 [40]. Thus, our study indicates that targeting the FAP cell population may be therapeutically explored in LGMD-R12 patients. However, while these findings are interesting and can provide direction for future therapeutic interventions in LGMD-R12 patients, deconvolution analysis should be interpreted carefully and further single cell analyses should be performed to quantitatively assess differences in cell type composition.

### 4.3. Different Muscles Express Different Gene Profiles in Healthy Controls

Our study demonstrated that three different muscles from the thigh in healthy control individuals show different gene expression profiles. Furthermore, the dystrophic process in LGMD-R12 patients resulted in different pathological effects on these three muscles. These basic molecular differences between muscles might at least partially explain the selective involvement of distinct muscles and the absent involvement of others in muscular dystrophies, such as LGMD-R12. The variation in gene expression profiles is far beyond the differences in the proportion of conventional fiber types I (slow-twitch) and IIA/B (fast-twitch), and occurs even in muscles with similar compositions of fiber types, such as the rectus femoris, vastus lateralis and semimembranosus muscles used in our study, which are all muscles with fiber type II predominance [78,79]. More recent studies showed that although skeletal muscle in humans consists of three distinct fiber types (I, IIa, IIx), muscle contains many more isoforms of myosin heavy and light chains, which are coded by a large number of different genes [51,52]. Stuart and coworkers demonstrated that specific human skeletal muscle fiber types contain different mixtures of myosin heavy and light chains, with fast-twitch (type IIx) fibers consistently containing myosin heavy chains 1 (MYH1), 2 (MYH2) and 4 (MYH4) and myosin light chain 1 (MYL1), and slow-twitch (type I) fibers always containing myosin heavy chains 6 (MYH6) and 7 (MYH7) and myosin light chain 3 (MYL3) [51]. In our study, we showed a clear gradient of fast- and slow-twitch muscle genes across the healthy muscles, with the rectus femoris showing the highest expression of the fast-twitch muscle myosins *MYH1*, *MYH2*, *MYH4* and *MYL1* and the semimembranosus presenting the highest expression of the slow-twitch markers *MYH6*, *MYH7* and *MYL3*. The vastus lateralis expressed intermediate levels of fast- and slow-twitch myosins. These differences in muscle fiber type isoforms might be one explanation of the differential involvement of muscles in muscular dystrophies, such as LGMD-R12.

Transcriptomic studies on different skeletal muscles in healthy animals also showed differential gene expressions between muscles. Terry et al. applied RNA-seq to profile RNA expression in 13 different skeletal muscles from mice and rats and showed extensive transcriptional diversity, with more than 50% of transcripts differentially expressed among skeletal muscles, an observation that cannot be explained by developmental history or fiber type composition alone [80]. The authors suggested a differential role of different transcription factors in distinct muscles to further explain their findings. Transcription factors recognize specific DNA sequences and control transcription by regulating the expression of genes in certain cells, at a certain time and in the right amount. Our study in humans demonstrated that several transcription factors were highly differentially expressed across the three healthy muscles, which may add to the explanation of the distinct susceptibility of different muscles in muscle diseases, such as LGMD-R12.

### 4.4. Conclusions and Future Perspectives

Overall, this study profiled the gene expression landscape in LGMD-R12. We identified baseline differences in expression levels between differently affected muscles and characterized the disease-associated changes. Our analysis showed differences in the cell composition of LGMD-R12 diseased and healthy muscle and suggested that transcription factor networks in specific cell populations underlie the differential predisposition of different muscles to degeneration. Because bulk RNA-sequencing only provides information on the average gene expression levels in muscle, we did not have the resolution to characterize these findings in more detail. Current single cell RNA-seq methodology would require a larger amount of muscle tissue per sample point, and was therefore not performed in this study. For further characterization at the cellular level, future studies using novel technologies such as single cell RNA-sequencing and possibly spatial RNA-sequencing need to be undertaken. Furthermore, this dataset can be used as a rich source for further exploration and referencing by the research community.

## Figures and Tables

**Figure 1 cells-11-01508-f001:**
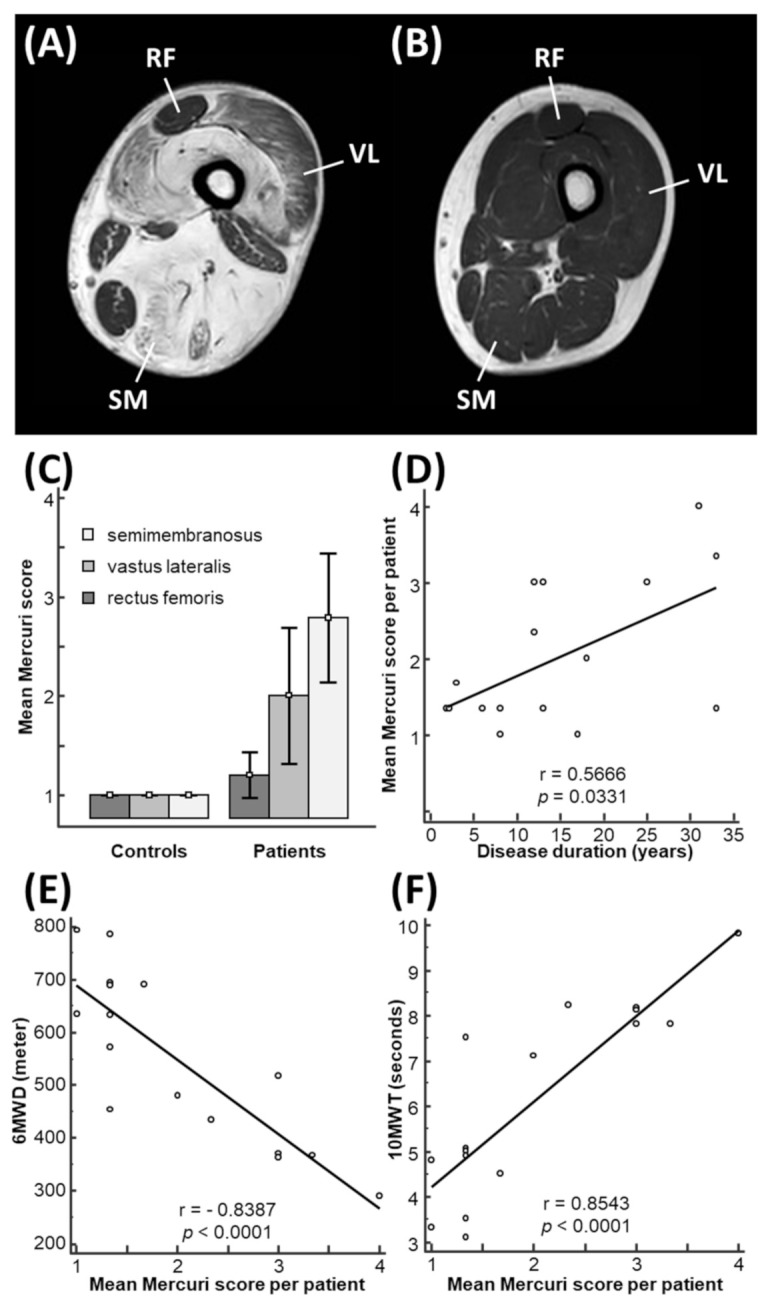
Mercuri score is correlated with functional outcomes in LGMD-R12 patients. (**A**) Axial MRI section through the left thigh of patient 6, indicating the differential involvement of the target muscles: rectus femoris (RF) with Mercuri score 1, vastus lateralis (VL) with Mercuri score 3 and semimembranosus (SM) with Mercuri score 4. (**B**) A similar axial MRI slice of a control individual is shown. All muscles in healthy controls are per inclusion criterion normal and thus have a Mercuri score of 1. (**C**) Mean Mercuri score for all patients and for all controls is shown for the three target muscles. The bars indicate the 95% confidence interval. (**D**) A significant correlation between the mean Mercuri score per patient (dots) and the disease duration is shown in a scatter diagram, with “r” indicating the Pearson correlation coefficient, and “*p*” the *p*-value. (**E**,**F**) Scatter diagrams illustrating a significant increase in the Mercuri score with decreasing 6MWD (**E**) and with increasing 10MWT (**F**).

**Figure 2 cells-11-01508-f002:**
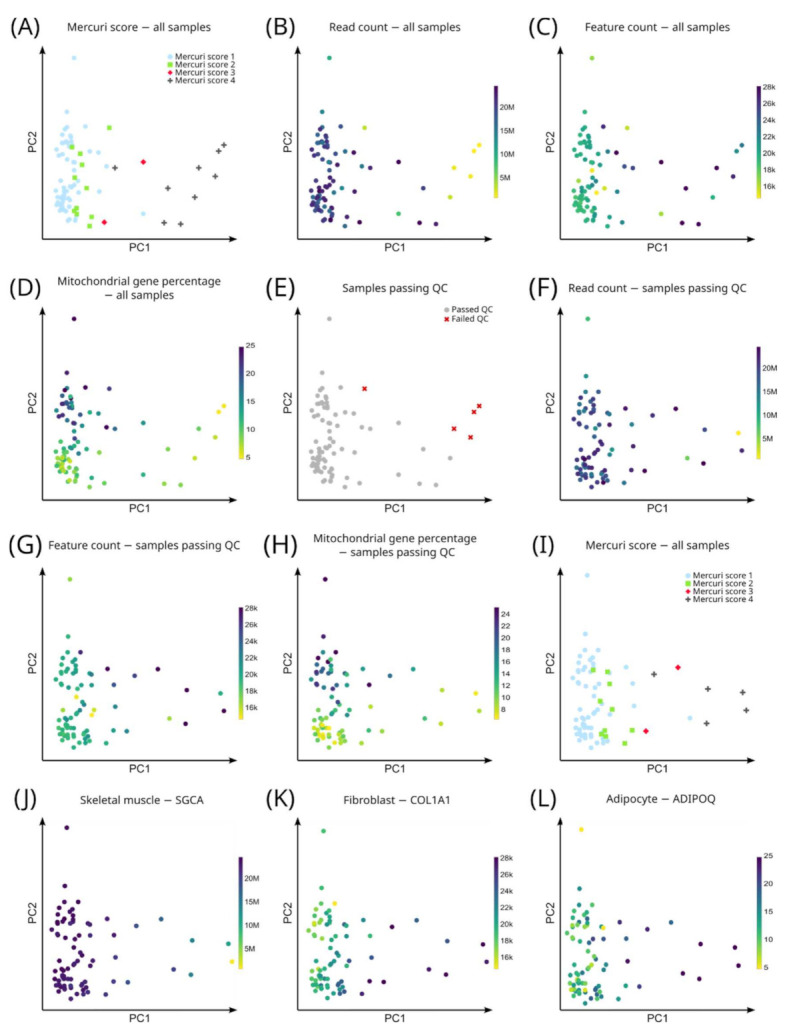
Unbiased transcriptome analysis of all genes. (**A**) PCA of all samples (n = 84), color coded for Mercuri score. (**B**) PCA of all samples (n = 84), color coded for read count. Note that samples with the highest Mercuri score had overall lower sequencing depth, which can be explained by the fact that the cellularity in high Mercuri score samples is lower due to replacement of muscle cells with fibrotic and adipose tissue. If there is low cellularity, then we also have lower transcripts and hence fewer reads that map to those transcripts. (**C**) PCA of all samples included in the study (n = 84), color coded for feature (gene) count. Note that samples with the highest Mercuri score had overall highest feature (gene) count. (**D**) PCA of all samples included in the study (n = 84), color coded for mitochondrial feature (gene) count. (**E**) PCA of all samples included in the study (n = 84), color coded for passing (grey, n = 79) or failing (red, n = 5) quality control (QC). All samples with 1M or more reads were considered to have passed QC. (**F**) PCA of samples that passed QC (n = 79), color coded for read count. (**G**) PCA of samples that passed QC (n = 79), color coded for feature (gene) count. (**H**) PCA of samples that passed QC (n = 79), color coded for mitochondrial feature (gene) count. (**I**) PCA of samples that passed QC (n = 79), color coded for Mercuri score. (**J**) PCA of samples that passed QC (n = 79), color coded for *SGCA* gene expression. Note, SGCA is a protein of the muscle membrane and SGCA gene expression (muscle membrane marker) was reduced in samples with higher Mercuri scores. (**K**) PCA of samples that passed QC (n = 79), color coded for *COL1A1* gene expression (fibroblast marker). (**L**) PCA of samples that passed QC (n = 79), color coded for *ADIPOQ* gene expression (adipocyte marker). In the PCA plots each sample is represented by a dot. The dots are placed on the graph in function of their gene signatures, the closer the dots are together, the more similar their gene expression profile is.

**Figure 3 cells-11-01508-f003:**
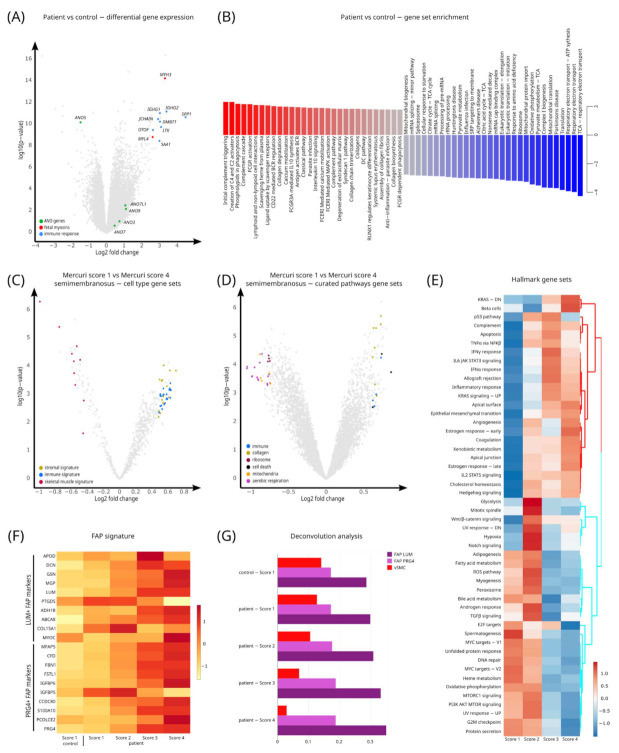
Gene signatures in limb-girdle muscular dystrophy R12. (**A**) Volcano plot of a differential gene expression analysis of LGMD-R12 patient versus control samples. Genes mentioned in text are highlighted in red. (**B**) Bar plot showing top enriched gene sets based on the results of differential gene expression analysis of LGMD-R12 patient versus control samples. (**C**) Volcano plot of a differential gene expression analysis of Mercuri score 1 versus Mercuri score 4 semimembranosus samples. Genes associated with immune cells, collagen production, ribosomes, cell death, mitochondria and aerobic respiration are color coded. Note that pathways associated with skeletal muscles are downregulated (red dots), whereas stromal (collagen) (green dots) and immune cell signatures (blue dots) are upregulated. (**D**) Volcano plot of a differential pathway expression analysis of Mercuri score 1 versus Mercuri score 4 semimembranosus samples. Pathways associated with skeletal muscle, stromal and immune cells are color coded. Note that genes associated with skeletal muscles are downregulated, stromal and immune cell signatures are upregulated. (**E**) Heatmap analysis of pathway activity associated with Mercuri score. Note that pathways associated with myogenesis and typical muscle signatures (oxidative phosphorylation, glycolysis and protein metabolism) are progressively downregulated, while pathways associated with cell death, inflammation and mesenchymal transition are upregulated. (**F**) Heatmap analysis of the proteoglycan 4+ (PRG4+) fibroadipogenic progenitor (FAP) and lumican+ (LUM+) FAP marker genes’ expression. These are the markers from the single cell analysis that were used as input for deconvolution analysis in panel (**G**). (**G**) Bar plot quantifying the deconvolution analysis predictions on the relative percentage of the indicated cell types.

**Figure 4 cells-11-01508-f004:**
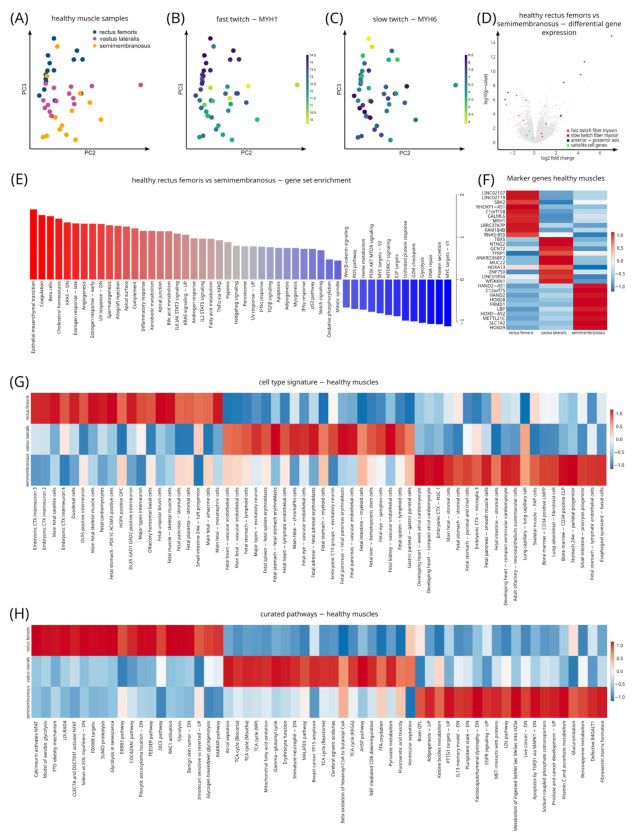
Gene signatures across muscles in healthy control muscles. (**A**) PCA of healthy control samples color coded for muscle of origin. (**B**) PCA of healthy control samples color coded for *MYH1* gene expression, which is characteristic for fast-twitch muscle fibers. Note the highest expression of *MYH1* in the rectus femoris muscle. (**C**) PCA of healthy control samples color coded for *MYH6* gene expression, which is characteristic for slow-twitch muscle fibers. Note the highest expression of *MYH6* in the semimembranosus muscle. (**D**) Volcano plot of a differential gene expression analysis of rectus femoris versus semimembranosus samples of control individuals. Genes associated with the indicated biological themes are color coded. (**E**) Bar plot showing top enriched gene sets based on the results of differential gene expression analysis of rectus femoris versus semimembranosus control samples. Note that the epithelial to mesenchymal transition signature is higher expressed in the semimembranosus muscle, while pathways associated with glucose metabolism, DNA repair and protein metabolism are higher in the rectus femoris. (**F**) Heatmap analysis of the top 10 marker genes of the biopsied muscles of healthy control individuals. (**G**) Heatmap analysis of the top 20 cell type pathways associated with each of the three muscles in healthy controls. Several interesting cell type signatures have been indicated. (**H**) Heatmap analysis of the top 20 biological processes associated with each of the three healthy muscles. Several interesting cell type signatures have been indicated.

**Table 1 cells-11-01508-t001:** Clinical and genetic features of the LGMD-R12 patients and MRI Mercuri scores of their biopsied muscles.

Patient Number	Gender	Age at Symptom Onset (y)	Age at Study Inclusion (y)	Disease Duration at Inclusion (y)	6MWD (m)	10MWT (s)	Mercuri Score SM	Mercuri Score VL	Mercuri Score RF	*ANO5* Mutations
1	M	30	63	33	366	7.8	4	4	2	c.191dupA (p.Asn64Lysfs*15); c.2317A>G (p.Met773Val)
2	M	31	33	2	785	3.5	2	1	1	c.191dupA (p.Asn64Lysfs*15); c.191dupA (p.Asn64Lysfs*15)
3	M	34	37	3	689	4.5	2	2	1	c.191dupA (p.Asn64Lysfs*15); c.191dupA (p.Asn64Lysfs*15)
4	M	30	38	8	632	4.8	1	1	1	c.191dupA (p.Asn64Lysfs*15); c.1961G>A (p.Arg654Gln) and c.155A>G (p.Asn52Ser)
5	M	47	59	12	369	8.1	4	4	1	c.172C>T (p.Arg58Trp); c.172C>T (p.Arg58Trp)
6	M	30	48	18	479	7.1	4	3	1	c.191dupA (p.Asn64Lysfs*15); c.692G>T (p.Gly231Val)
7	M	38	55	17	792	3.3	2	1	1	c.1213C>T (p.Gln405X); c.1733T>C (p.Phe578Ser)
8	M	39	64	25	362	8.1	4	4	2	c.191dupA (p.Asn64Lysfs*15); c.191dupA (p.Asn64Lysfs*15)
9	M	35	43	8	452	7.5	2	1	1	c.1210C>T (p.Arg404X); c.2387C>T (p.Ser796Leu)
10	M	33	46	13	516	7.8	4	3	2	c.191dupA (p.Asn64Lysfs*15); c.294+1G>A (p.?)
11	M	15	48	33	630	5.0	2	1	1	c.649-2A>G (p.?); c.679-2A>G (p.?)
12	M	33	64	31	288	9.8	4	4	4	c.41-1G>A (p.?); c.752C>T (p.Pro251Leu)
13	M	13	26	13	570	5.0	2	1	1	c.191dupA (p.Asn64Lysfs*15); c.191dupA (p.Asn64Lysfs*15)
14	M	34	36	2	692	4.9	2	1	1	c.191dupA (p.Asn64Lysfs*15); c.242A>G (p.Asp81Gly)
15	M	28	40	12	433	8.2	4	2	1	c.191dupA (p.Asn64Lysfs*15); c.1213C>T (p.Gln405X)
16(#)	M	25	31	6	687	3.1	2	1	1	c.2411G>C (p.Cys804Ser); c.1627dupA (p.Met543Asnfs*11)

y, years; 6MWD, six-minute walk distance; m, meter; 10MWT, 10-m walk test; s, seconds; SM, semimembranosus muscle; VL, vastus lateralis muscle; RF, rectus femoris muscle; *ANO5*, anoctamin-5 gene; M, male; 16(#), patient 16 was not included for RNA-seq because there was no RNA yield in the three muscle biopsies of this patient. The reference sequence to which the reported variants are referred is NM_213599.2.

**Table 2 cells-11-01508-t002:** Top 50 of most differentially expressed genes across healthy muscles.

Marker Gene Analysis across Healthy Muscles			
Rank	Semimembranosus	Rectus Femoris	Vastus Lateralis
1	HAND2-AS1	TBX5	LINC02107
2	C12orf75	NTNG2	LINC02119
3	HAND2	GCNT2	SBK2
4	HOXD8	TYRP1	RHOXF1-AS1
5	FRMD1	ANKRD36BP2	C1orf158
6	LBP	MUC22	CALML6
7	HOXD-AS2	HOXA13	MYH1
8	METTL21C	ZNF750	LRRC37A7P
9	SLC1A2	LINC01854	FAM184B
10	HOXD9	WFIKKN1	RNA5-8S5
11	IL31RA	TBX5-AS1	LINC01886
12	KIF1A	FNDC10	FEZF1-AS1
13	ANGPTL8	IPCEF1	CRNDE
14	CFAP57	JCHAIN	CPXM1
15	DNAH3	CD300LB	TCF24
16	IL22RA1	SATB2-AS1	CDH22
17	BDNF	TUBB1	PAX3
18	LRRC52	DIRAS1	SNORD115-30
19	C10orf67	SPTA1	GGT7
20	CAPN8	LINC01968	MYHAS
21	CROCC2	HMGCS2	RGS10
22	LAD1	TREM1	SLITRK3
23	ANKRD18B	CCDC189	PLCH1
24	GLYAT	COL9A1	IGFN1
25	SCD	P2RX3	ATRNL1
26	IRX6	RRM2	ACTN3
27	PAQR9-AS1	IGHM	SHISA2
28	LINC01484	IGHV3-7	GADD45G
29	C6orf132	SPAG17	MIR503HG
30	HOXD3	RPS27AP9	FGF10
31	TMC1	PAX1	GREM2
32	FOS	IRX4	GDA
33	SCRT1	CXCR2P1	OPRD1
34	RNY4P10	DNAH11	RN7SL813P
35	SLC26A9	SIM2	UBASH3B
36	CCDC78	TMEM163	SNORD115-23
37	COMP	CDH20	GDNF
38	ADRB1	TMEM105	NANOS1
39	PLPPR1	SKA3	LINC01773
40	GPR39	SLC7A11-AS1	HSD52
41	LINC00877	EPB42	NPR3
42	SLC29A4	SLC30A8	NME9
43	RSPO1	HS6ST2	CHAD
44	BARX2	FGD5P1	MKRN3
45	LOXL1-AS1	SEL1L2	SCT
46	GPA33	NLRP12	RN7SL267P
47	SDR42E2	SLC4A10	KHDRBS2
48	CERS3	CD160	MYH4
49	C2CD4B	ETF1P2	RN7SL541P
50	TRPM1	RYR3	SH2D1B

**Table 3 cells-11-01508-t003:** Top 50 of most differentially expressed genes across LGMD-R12 patient muscles.

Marker Gene Analysis across Patient Muscles			
Rank	Semimembranosus	Rectus Femoris	Vastus Lateralis
1	COMP	LINC02107	SIM2
2	HAND2	MYH1	P2RX3
3	HAND2-AS1	LRRC37A7P	HMGCS2
4	COL20A1	AQP4	CHAC1
5	AQP6	LINC01773	LINC01854
6	ADIPOQ	LINC02119	KCTD8
7	PLEKHG4B	C1orf158	TBX5-AS1
8	FRMD1	PVALB	ZNF750
9	CIDEC	ACTN3	IGLV3-21
10	TNMD	CALML6	IGLC3
11	HYDIN	MYLK4	HOXA13
12	SCD	FBP2	IGHD
13	SALL1	HCN1	ADAMTS19-AS1
14	LRRC74A	UNC13C	LAMC3
15	LEP	B3GALT1	MAPT-AS1
16	SLC1A6	ERBB4	FBP2
17	PLA2G2A	GREM2	HAND2-AS1
18	KCNQ2	ATP2A1	NEU4
19	LGALS12	RHOXF1-AS1	SNCB
20	CHI3L1	LINC01886	IL20RA
21	GRM5	NANOS1	AQP4
22	SLC5A10	SHISA2	HOXC12
23	CCL18	UGT3A1	DIRAS1
24	CUX2	MLF1	SLC16A3
25	KLB	NRG4	TSHR
26	MUC16	SH2D1B	GDNF
27	GRIN2B	PLCH1	KLHDC7B
28	PIEZO2	LRRC3B	LINC01018
29	SAA1	MYHAS	SLC51A
30	MKX	MYLK2	GHRL
31	GRM4	FEZF1-AS1	TBX1
32	TSPEAR	FAM166B	IGHV1-3
33	TNC	SNORD23	TYRP1
34	MYEOV	ENO3	CRYM
35	PCK1	ENSAP2	PIANP
36	PLIN1	IRX3	IGKV1-5
37	CACNA1I	CDH22	TMEM26
38	USH2A	ATRNL1	MTND4P24
39	MUC6	LANCL1-AS1	RN7SKP276
40	COL22A1	NEK10	FAM166B
41	SCUBE1	PDE4DIPP1	HPN
42	SCRG1	TMEM266	OSCAR
43	S100A3	FABP7	HES7
44	KRT7	KLHL38	PAX1
45	OPRM1	AGMAT	ASB12
46	DUX4L19	SMCO1	DDX11L2
47	MYBL2	ASB14	ANKRD20A21P
48	AMZ1	NPSR1-AS1	SPAG17
49	RASAL1	PITX1	TNNI3
50	MROH4P	DDIT4L	C1orf105

## Data Availability

Data supporting reported results can be obtained from the corresponding author on reasonable request. RNA-seq datasets will be deposited in the GEO database.

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
