# Peer review of "Unraveling the Molecular Basis of the Dystrophic Process in Limb-Girdle Muscular Dystrophy LGMD-R12 by Differential Gene Expression Profiles in Diseased and Healthy Muscles"

_cells, 2022, doi:10.3390/cells11091508_

Round 1
Reviewer 1 Report
The manuscript by Depuydt et al. presents a comparative transcriptome study of three different upper leg muscles in 16 male LGMD-R12 21 patients and 15 age-matched healthy male control individuals.
The strength of the study is the high number of study subjects, their detailed clinical and genetic description, and notably the fact of having obtained three biopsies of three defined muscles from each patient at the same time point. The authors used MRI to grade the state of muscular dystrophy by quantifying fatty-fibrotic changes in SM, VL and RF muscles. They first demonstrated the correlation of MRI changes with clinical progression, thus qualifying MRI changes as surrogate for clinical state in this patient cohort. The authors performed bulk RNA-Seq on the muscle biopsies. They show that the transcriptome profile well correlates with the degree of dystrophic remodeling of muscle tissue seen by MRI. The authors also compared the transcriptome between the three different healthy muscle, which highlights the specific molecular signature of the different muscles.
One weakness of the study is the choice of methodology. The choice of bulk RNA Seq informs mainly about expected dystrophic changes, which in my opinion adds little information as already known from classical histology. It does not allow to study transcriptional changes within cell populations, which would be particularly interesting to explore underlying pathophysiological mechanisms. Authors missed the opportunity of performing snRNA Seq or spatial transcriptomics. Therefore, no information were obtained on muscle fiber compartmentalization and myonuclear specification, differential analysis of inflammatory infiltrates, mononuclear components (e.g. muscle stem cells) etc.. In consequence, the results contribute little to the main scientific question of this work: what is molecular mechanism driving the different disease progression in the three analyzed muscles? Authors should discuss the limits of employed methodology. They also should reason of why employing more advanced techniques, e.g. snRNA Seq, was inconvenient or even impossible. I could imagine that low sample size following needle biopsy was a limiting factor (current snRNA Seq requires about 100 mg muscle tissue per sample point).
The other weakness is the little ambitious bioinformatic analysis of the RNA Seq data set. Authors show that number of reads of marker genes of tissue changes correlate with MRI surrogate. More in-depth analysis could have been performed such as pathway, gene network and functional annotation analysis allowing for functional clustering of highest enrichment scores for gene ontology as well as depiction in clustergrams of genes associated with GO terms. Such more detailed analysis would be notably interesting for comparing the control samples.
In conclusion, it the added value of performing bulk-RNA Seq for diagnostic purpose or clinical research remains little convincing.
Reviewer 2 Report
The study presented in this paper provides new relevant evidences to understand the molecular mechanisms underlying muscle degeneration in limb-girdle muscular dystrophy R12.
I found it well designer and clearly presented, thus I have one minor criticism, that is to specify the reference sequence to which the reported variants are referred (i.e.: NM_213599.3? Please check).
Reviewer 3 Report
The manuscript by Depuydt et al, presents a study where they aim to identify genes and pathways that underlie LGMD-R12 and the various types of muscle affected by the disease through the technique of transcriptomics. They identified several genes that were differentially expressed as well as differently affected muscle types. The study thus provides hallmark fingerprints that could further be therapeutically explored to address the concerns of the disease. Further functional assays will be warranted to provide conclusive findings. However, I think this study will be very relevant and significant to the readers of Cells and, also to the scientific community in general.
Minor comments:
Grammatic correction required in Line 199
In fig. 1, what could be the difference in patients that deviates from the relation between the Mercuri score and the disease duration. That is, patients that show a long disease duration, but otherwise show a lower mercuri score?
Fig 2. Why do patients with high Mercuri score have a low sequencing depth?
Reviewer 4 Report
The Authors show transcriptomic data performed on three thigh muscles in 16 male affected by LGMD-R12 anoctamin5 related, and 15 age-matched male controls.
The Authors showed that LGMD-R12 muscular dystrophy is associated with an up-regulation of gene indicative of fibroblast and adipocyte replacement, and immune cell infiltration, while gene signatures associated with striated muscle (protein synthesis, OXPHOS, glycogen-, glucose-, and amino acid metabolism) are downregulated in dystrophic muscle.
They also clearly correlated the severity of radiological involvement (measured by the MRC scale) with the gene expression of genes associated with muscle injury and inflammation, as well as genes involved in muscle repair/regeneration.
Analysis of baseline differences in between muscles in healthy individuals indicated that muscles that are the most affected by LGMD-R12 have the lowest expression of transcription factor networks involved in muscle regeneration and satellite cell activation.
The aim of the project is well defined, the results are clearly explained and the conclusions consistent with the results. Moreover, the novelty of the project is high, and the work provides new insights on LGMD-R12 pathophysiology.
Minor revisions:
- Line 42: please change LGMD-R12 with LGMD-R12 anoctamin5 related (LGMD R12)
- Line 55: inflammatory infiltrates are also present in muscle biopsies from LGMDR12 patients as reported in the discussion (Silva AMS et al. Clinical and molecular findings in a cohort of ANO5-related myopathy. Ann Clin Transl Neurol. 2019 Jul;6(7):1225-1238. doi: 10.1002/acn3.50801. Epub 2019 Jun 11)
- Line 518: The involvement of immune response in Muscular Dystrophies is reviewed in “Tidball JG et al. Immunobiology of Inherited Muscular Dystrophies. Compr Physiol. 2018 Sep 14;8(4):1313-1356. doi: 10.1002/cphy.c170052. “
- Line 518: Inflammatory features are also reported in sarcoglycanopathies: “Panicucci C et al. Muscle inflammatory pattern in alpha- and gamma-sarcoglycanopathies. Clin Neuropathol. 2021 Nov-Dec;40(6):310-318. doi: 10.5414/NP301393”.
Round 2
Reviewer 2 Report
The paper can be accepted in the present form
Reviewer 3 Report
The authors have addressed all comments satisfactorily.
Reviewer 4 Report
I agree with publication of the article in the present form.